

# Intein-mediated backbone cyclization of entolimod confers enhanced radioprotective activity in mouse models

Bingyu Ye[1,2,3], Wenlong Shen[3], Minglei Shi[3], Yan Zhang[3], Cunshuan Xu[1,2] and Zhihu Zhao[3]

[1] College of Life Science, Henan Normal University, Xinxiang, China
[2] State Key Laboratory Cultivation Base for Cell Differentiation Regulation, College of Life Science, Henan Normal University, Xinxiang, China
[3] Beijing Institute of Biotechnology, Beijing, China

## ABSTRACT

**Background:** Entolimod is a *Salmonella enterica* flagellin derivate. Previous work has demonstrated that entolimod effectively protects mice and non-human primates from ionizing radiation. However, it caused a "flu-like" syndrome after radioprotective and anticancer clinical application, indicating some type of immunogenicity and toxicity. Cyclization is commonly used to improve the in vivo stability and activity of peptides and proteins.

**Methods:** We designed and constructed cyclic entolimod using split *Nostoc punctiforme* DnaE intein with almost 100% cyclization efficiency. We adopted different strategies to purify the linear and circular entolimod due to their different topologies. Both of linear and circular entolimod were first purified by Ni-chelating affinity chromatography, and then the linear and circular entolimod were purified by size-exclusion and ion-exchange chromatography, respectively.

**Results:** The circular entolimod showed significantly increased both the in vitro NF-κB signaling and in vivo radioprotective activity in mice.

**Conclusion:** Our data indicates that circular entolimod might be a good candidate for further clinical investigation.

## INTRODUCTION

Entolimod (previously called CBLB502) is a truncated derivative of the *Salmonella* flagellin protein. It is substantially less immunogenic than flagellin but retains its TLR5-dependent NF-κB-inducing activity and radioprotective capability (*Burdelya et al., 2008*). Entolimod is currently under development as a medical radiation countermeasure under the FDA's Animal Efficacy Rule and has demonstrated efficacy for both reducing damage to radiosensitive hematopoietic (HP) and gastrointestinal tissues as well as in improving their regeneration (*Krivokrysenko et al., 2012*). Moreover, entolimod is also shown to be an effective antitumor agent in several in vivo models (*Burdelya et al., 2012*; *Cai et al., 2011*). Entolimod is now considered for clinical use because unlike activation of some other toll-like receptors (TLRs), the specific profile of cytokines

Corresponding authors
Cunshuan Xu, 003009@htu.edu.cn
Zhihu Zhao, zhaozh@bmi.ac.cn

induced following TLR5 stimulation by entolimod does not lead to septic shock-like syndrome or a "cytokine storm" of IL-1 and tumor necrosis factor (TNF) (*Akira & Takeda, 2004*; *Carvalho et al., 2011*; *Vijay-Kumar et al., 2008*). Entolimod thus has prospective clinical applications as a radioprotective and anticancer agent; however, being a flagellin variant, it can still cause a "flu-like" syndrome after injection, indicating some type of immunogenicity and toxicity (*Ding et al., 2012*; *Burdelya et al., 2013*; *Hossain et al., 2014*; *Kojouharov et al., 2014*; *Yang et al., 2016*; *Brackett et al., 2016*). Another concern is that the time frame for effective entolimod administration is relatively narrow, especially at very high doses of radiation exposure (10 or 13 Gy, for instance) (*Burdelya et al., 2008*). Therefore, it is important to develop entolimod derivates with increased activity, stability, and more efficient recombinant production.

Intein-mediated backbone cyclization of proteins is a widely used approach for improving protein stability and biological activity (*Tarasava & Freisinger, 2014*; *Iwai & Plückthun, 1999*); therefore, it is now considered a powerful tool for enhancing the efficacy of protein-based therapeutics (*Tavassoli, 2017*). Studies have shown that relative to the linear protein, the cyclized protein is not susceptible to hydrolysis by exogenous proteases owing to its conformational rigidity arising from lack of both amino and carboxyl termini (*Iwai & Plückthun, 1999*; *Evans, Benner & Xu, 1999*; *Horton, Bourne & Smythe, 2002*). Both expressed protein ligation (*Camarero & Muir, 1999*) and protein *trans*-splicing (PTS) (*Scott et al., 1999*) have been used to generate cyclic peptides and proteins. An alternative complementary method termed as split-intein mediated circular ligation of peptides and proteins (SICLOPPS) has also been developed to facilitate the convenient and efficient use of intein splicing (*Scott et al., 1999*). The SICLOPPS construct contains three parts: the C-terminal intein domain, target sequence, and N-terminal intein domain. In this construct, the target sequence can be head-to-tail cyclized. SICLOPPS originally used the naturally occurring split intein DnaE from *Synechocystis sp.* PCC6803 (*Ssp*) (*Wu, Hu & Liu, 1998*), but recently this intein has been replaced with the faster splicing engineered intein from *Nostoc punctiforme* (*Npu*), which is also significantly more tolerant of amino acid diversity in the extein sequence (*Townend & Tavassoli, 2016*). SICLOPPS is mainly used to generate cyclic peptide libraries in drug discovery. For example, researchers have discovered cyclic peptide inhibitors of DAM methyltransferase (*Naumann, Tavassoli & Benkovic, 2008*), ClpXP protease (*Cheng et al., 2007*), hypoxia-inducible factor-1, a variety of protein–protein interactions (*Miranda et al., 2013*), and inhibitors that reduce the toxicity of α-synuclein, a key protein in Parkinson's disease (*Kritzer et al., 2009*). Cyclization of TEM-1β-lactamase, GFP and VP1 proteins has shown increased thermostability than the linear form and significantly more resistance to proteolysis by exopeptidase (*Iwai & Plückthun, 1999*; *Iwai, Lingel & Pluckthun, 2001*; *Zhao et al., 2010*; *Qi & Xiong, 2017*). However, the use of circular proteins as new and potential drug targets to prevent and treat diseases has not been explored.

In this work, based on PTS, we used the split intein DnaE from *Npu* to generate a split functional N- and C-terminal intein ($I_C$ and $I_N$) to cyclize the entolimod protein in *E. coli*. The cyclization reaction was achieved by sandwiching the entolimod with Strep II-tag, His$_6$-tag between the C-terminal domain (*Npu*$I_C$) and N-terminal domain

($Npu\text{I}_\text{N}$) of the $Npu$ DnaE intein in the order of $Npu\text{I}_\text{C}$—Strep/entolimod/His$_6$—$Npu\text{I}_\text{N}$. This design was successfully cloned, and expressed the entolimod protein in circular form, in which the N- and C-termini are covalently joined by an amide bond with approximately 100% cyclization efficiency. After expression, we found that the linear and circular form of entolimod required different purification strategies owing to their different topologies. After purification, we compared their biological activity both in vitro and in vivo; cyclic entolimod (hereafter referred as circ-entolimod) showed significant activity in both in vitro a dual-luciferase reporter assay and in vivo radioprotective activity in C57BL/6 mouse models. Because of the excellent biological activity of circ-entolimod, we could achieve the same effect as that of linear entolimod (or lin-entolimod) by administering a much lower dose of circ-entolimod, which could further reduce its potential immunogenicity and toxicity.

## MATERIALS AND METHODS

### Cloning and construction of lin- and circ-entolimod expression plasmids

The full-length flagellin gene was amplified by forward primer 1 carrying the $Nde$I restriction site and StrepII-tag (Table 1, bold font) and reverse primer 1 with the $Hin$dIII site and His$_6$-tag (Table 1, bold font) from *Salmonella enterica* serovar Dublin genome, as previously described (*Burdelya et al., 2008*). We used the flagellin gene as a template and amplified the entolimod N-terminal using forward primer 1 and reverse primer 2, and the C-terminal using forward primer 2 and reverse primer 1. The PCR products of the entolimod N-terminal and C-terminal were mixed and used as templates to amplify entolimod using forward primer 1 and reverse primer 1. The sequence 5′-TCCCCGGGAATTTCCGGTGGTGGTGGTGGAATTCTAGACTCCATGGGT-3′ was the linker between entolimod N-terminal and C-terminal. The resulting DNA segment was ligated into pET-28a(+) (Novagen #69864-3), resulting in pET/entolimod-28a(+). We then used the same strategies to clone $Ssp$ DnaE intein and $Npu$ DnaE intein into pET-28a(+). We amplified $Ssp$ I$_\text{C}$ using forward primer 3 carrying an $Nco$I restriction site and reverse primer 3 carrying a $Bam$HI site from pSFBAD09 (Addgene # 11963, *Züger & Iwai, 2005*), the product was ligated into pET-28a(+), to generate pET/$Ssp$ I$_\text{C}$-28a(+), the italicized base in reverse primer 3 contain a linker sequence with $Kpn$I, $Bam$HI, and $Nde$I cloning sites and three different termination codon frames TAA, TGA, and TAG. $Ssp$ I$_\text{N}$ was amplified using forward primer 4 with a $Bam$HI site and reverse primer 4 with a $Hin$dIII site from pJJDuet30 (Addgene # 11962, *Züger & Iwai, 2005*), and then ligated into pET/$Ssp$ I$_\text{C}$-28a(+) to produce pET/$Ssp$ DnaE-28a(+). We obtained pET/$Npu$ I$_\text{C}$-28a(+) using forward primer 5 and reverse primer 5 with the same restriction enzymes as those used for pET/$Ssp$ I$_\text{C}$-28a(+) from pSKBAD02 (Addgene # 15335, *Iwai et al., 2006*), and pET/$Npu$ DnaE-28a(+) was constructed using forward primer 6 and reverse primer 6 with the same restriction enzymes as pET/$Ssp$ DnaE-28a(+) from pSKDuet01 (Addgene # 12172, *Iwai et al., 2006*). Lastly, we used forward primer 1 and primer 7 carrying a $Bam$HI site, His$_6$-tag (Table 1, bold font) to amplify entolimod from

**Table 1 Primer sequences used in this study.**

| Primer | Sequence (5′–3′) |
| --- | --- |
| Forward primer 1 | AAA**CATATG**T**TGGAGCCACCCGCAGTTCGAAAAA**GCACAAGTCATTAATACAAACAG |
| Reverse primer 1 | GCG**AAGCTT**G**TGATGATGATGATGATG**TCATTAACGCAGTAAAGAGAGGACGTT |
| Forward primer 2 | TAGAAACGCGATCGATTTCTTCCAGACGTTGCTGAATTTCATCCTGGATAG |
| Reverse primer 2 | GGTGGTGGTGGTGGAATTCTAGACTCCATGGGTACATTAATCAATGAAGAC |
| Forward primer 3 | TAA**CCATGGG**CGTTAAAGTTATCGGTC |
| Reverse primer 3 | TAA*GGATCC*GGTACC*TACTCAGTTACATATG*ATTGAAACAATTTGCAGC |
| Forward primer 4 | AAA**GGATCC**TGCTTAAGTTTCGGTACT |
| Reverse primer 4 | ATG**AAGCTTT**CATTATTTGATAGTACCAGCGTCC |
| Forward primer 5 | TAA**CCATGGG**CATCAAAATAGCCACACGTAAAT |
| Reverse primer 5 | TAA*GGATCC*GGTACCTACTCAGTTACATATG*ATTGAAACAATTAGAAG |
| Forward primer 6 | ACA**GGATCC**TGTTTAAGCTATGAAACGGAAATATTG |
| Reverse primer 6 | ATG**AAGCTTT**TATCA ATTCGGCAAATTATCAACCCG |
| Primer 7 | AAA**GGATCC**GTGATGATGATGATGATGACGCAGTAAAGAGAGGACGT |

pET/entolimod-28a(+), and then produce pET/*Ssp* DnaE-28a(+) and pET/*Npu* DnaE-28a (+) to generate the final circ-entolimod expression plasmids pET/*Ssp* DnaE/entolimod-28a(+) and pET/*Npu* DnaE/entolimod-28a(+).

## Expression of recombinant entolimod proteins and cell lysis

The expression plasmids pET/entolimod-28a(+), pET/*Ssp* DnaE/entolimod-28a(+) and pET/*Npu* DnaE/entolimod-28a(+) were transformed into *E. coli* BL21(DE3) (Tiangen Biotech, Beijing, China). The cells were grown overnight at 37 °C in LB medium supplemented with 50 μg/mL kanamycin. Protein overexpression was induced by adding isopropyl β-D-1-thiogalactopyranoside (IPTG) to a final concentration of 0.5 mM in the bacterial culture (when optical density at 600 nm reached ∼0.5) and the incubation was extended for additional 3–5 h at 37 °C. The cells containing the expression plasmids pET/entolimod-28a(+), pET/*Ssp* DnaE/entolimod-28a(+), and pET/*Npu* DnaE/entolimod-28a(+) were subsequently harvested at 6,000×*g* for 25 min. After centrifugation, the three cell pellets were respectively sonicated (30% amplitude for 30 min, 5 s on, 5 s off) in lysis buffer: 20 mM Tris, 500 mM NaCl, 10 mM Imidazole, pH 8.0. The target proteins expressed as inclusion bodies (∼80% of total) were confirmed by SDS–PAGE analysis.

## Purification of lin- and circ-entolimod proteins

The lin-entolimod was purified described previously (*Burdelya et al., 2008*). First, the lin-entolimod-expressing cells were suspended in lysis buffer comprising 20 mM Tris, 500 mM NaCl, and 10 mM Imidazole, pH 8.0, followed by sonication in an ice water bath to minimize thermal damage to proteins. The supernatant and pellet were then collected. The pellet was used for further purification and could be completely solubilized in Buffer A (20 mM Tris, 500 mM NaCl, 2 M Urea, pH 8.0). Then, the lin-entolimod was further purified by Ni-chelating affinity column chromatography (GE Healthcare, Little Chalfont, UK) and could be readily eluted in Buffer B (20 mM Tris, 500 mM NaCl, 250 mM Imidazole, pH 8.0).

We further purified lin-entolimod according to the procedure described previously, and performed desalting followed by size-exclusion chromatography (GE Healthcare, Little Chalfont, UK) (*Burdelya et al., 2008*). However, the same purification methods were not suitable for circ-entolimod. Like lin-entolimod, circ-entolimod was also completely solubilized in Buffer A as described above. The first one-step elution of lin-entolimod (Ni-chelating affinity column chromatography) was not suitable for circ-entolimod purification. Therefore, we increased the concentration of imidazole (0–500 mM) in a step-wise manner to elute circ-entolimod (three imidazole gradients: 50, 200, and 500 mM). In this step, an alternative strategy for purifying circ-entolimod (or lin-entolimod) is Strep-Tactin affinity purification system (IBA Lifesciences, Goettingen, Germany) (according to manufacture's instructions). After purification by Ni-chelating affinity chromatography, the total sample was desalted (20 mM phosphate buffer (PB), pH 7.4) using a G25 column (GE Healthcare, Little Chalfont, UK). After desalting, ion-exchange chromatography (IEC) was performed on a column of HiTrap Q HP (anion exchange) resin (GE Healthcare, Little Chalfont, UK). We used 20 mM PB, pH 7.4 as the ion-exchange binding buffer. The circ-entolimod was then eluted using an increasing salt (10 mM–1 M NaCl) gradient. Protein concentration was measured by a BCA Protein Assay Kit (Tiangen Biotech, Beijing, China) and protein fractions were analyzed by SDS–PAGE stained with Coomassie blue. Endotoxin was measured by limulus lysate agent. The buffers of purified lin-entolimod and circ-entolimod were changed to phosphate-buffered saline (PBS) and the proteins were stored at −80 °C until used.

## MALDI-TOF imaging mass spectrometry

Mass spectrometer (MS) was performed on an ultrafleXtreme MALDI-TOF/TOF MS instrument (Bruker Daltonics, Billerica, MA, USA). The sample solution is mixed in equal volumes with matrix solution (containing 4 mg/ml CHCA, 80%CAN and 0.1%TFA), and then take 1 $\mu$l of the mixed sample point to the target for dry detection. The MS parameters were as follows, mode: LP 5–50 KDa, *m/z* range 10 to 60 K, 500 laser shots per spectrum, ion source voltage 1 set at 20 kV, ion source voltage 2 set at 18.40 kV, lens voltage set at 6.2 kV.

## NF-κB-dependent dual-luciferase reporter assays

Human embryonic kidney cells 293 (HEK293) (ATCC # CRL-1573) at 70–90% confluence were transfected with plasmids pNF-κB-Luc (#219078; Agilent, Santa Clara, CA, USA) and pRL-SV40 (#E2231; Promega, Madison, WI, USA) using Lipofectamine™ 2000 (#11668019; ThermoFisher Scientific, Waltham, MA, USA) as per the manufacturer's protocol. The lin- and circ-entolimod were added to the cells at different concentrations (0.00001, 0.001, and 0.1 $\mu$M) at 24 h after transfection. Firefly and *Renilla* luciferase assays were performed 6 h after addition of lin- and circ-entolimod to the culture medium using the dual-luciferase reporter assay system (#E1980; Promega, Madison, WI, USA).

## Irradiation and radioprotection of mice

C57BL/6 mice were obtained from Vital River Laboratories (VRL) (Beijing, China). All animal experiments were performed according to the Guide for the Care and Use of Medical Laboratory Animals (Ministry of Health, China, 1998) and with the ethical

approval of the Beijing Institute of Biotechnology. The approval documentation number is No.545 ([2001]). Male mice, 6–8 weeks old, weighing 18–22 g were used for the experiment (12 animals per group, six groups, a total of 72 animals). The mice were divided into three groups with similar weight distribution: PBS (control) group, lin-entolimod treatment group, and circ-entolimod treatment group. The mice were subjected to total body irradiation (TBI) with $^{60}$Co-gamma to a total dose of 9 and 14 Gy at a dose rate of 153.33 cGy/min (Institute of Radiation Medicine, Beijing, China). Mice were irradiated on a rotating platform to ensure even dose delivery to all tissues. The lin- and circ-entolimod were injected subcutaneously (s.c.) 30 min before TBI at a dose of 0.2 mg/kg. The PBS group was injected an equivalent volume of PBS instead. The survival rate of mice was observed 30 days after TBI.

### Statistical analysis

Each column is presented as means ± SD of three independent experiments. The results were statistically evaluated for significance using the Student's $t$-test. The survival rate was statistically evaluated for significance using the log-rank test. $P < 0.05$ was considered statistically significant.

## RESULTS

### *Npu* DnaE split intein-mediated entolimod cyclization

According to the principle of PTS, the two fragments of intein, $I_N$ and $I_C$, interact with each other to form an active intein that splices to cyclize the proteins or peptides placed in between. In this case, the entire entolimod gene was sandwiched between the $I_C$ and $I_N$ of *Ssp* or *Npu* intein with CFN residues at the C terminus from the native C-terminal extein sequence of DnaE intein, and HM and GS residues at its C terminus and N terminus due to cloning (Figs. 1A–1C).

The circ-entolimod was readily apparent by SDS–PAGE upon IPTG induction under the expression plasmids pET/*Npu* DnaE/entolimod-28a(+). Figure 2A shows the expression and the expected splice product circ-entolimod (~33.7 kDa); the intein ($I_N$ and $I_C$) band was clearly observed in two fragments, $I_N$ (~11.9 kDa) and $I_C$ (~4.1 kDa), after splicing. The circ-entolimod migrated more rapidly in SDS–PAGE analyses than did lin-entolimod (Fig. S1), implying an additional topological constraint (*Iwai & Plückthun, 1999*; *Zhao et al., 2010*; *Qi & Xiong, 2017*). As reported previously, lin-entolimod is mainly expressed in the form of inclusion bodies (~80%), but is resistant to thermal denaturation at 90 °C for 20 min and does not need refolding (*Burdelya et al., 2008*). In our study, after sonication, circ-entolimod also predominantly formed inclusion bodies (Fig. 2A). We then tested whether entolimod cyclization could be mediated by *Ssp* intein. Unfortunately, the results indicated that pET/*Ssp* DnaE/entolimod-28a(+) failed to induce the expression of circ-entolimod, and the precursor (*Ssp* DnaE-entolimod) was clearly observed in the gel (Fig. 2B). Therefore, *Npu* intein was considered a better choice for entolimod cyclization. Previous studies have shown that *Npu* intein is more tolerant of amino acid substitutions in the C-terminal extein sequence than *Ssp* intein (*Townend & Tavassoli, 2016*), and also has exhibited the highest efficiency for

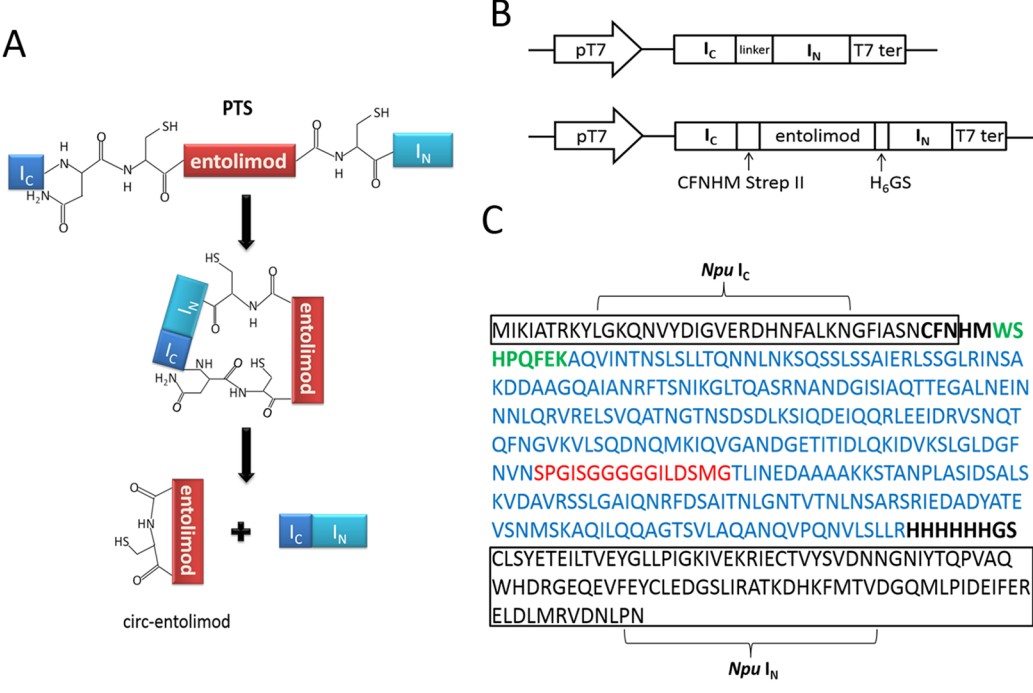

**Figure 1** **Entolimod cyclization in vivo using trans-splicing activity of *Npu* DnaE intein.** (A) The two fragments of intein, $I_N$ and $I_C$, interact to form an active intein that splices to cyclize entolimod placed in between. Splicing mediates the ligation of the N and C termini of entolimod through a native peptide bond. (B) Schematic representation of expression plasmids pET/*Ssp* DnaE-28a(+) or pET/*Npu* DnaE-28a(+) and pET/*Ssp* DnaE/entolimod-28a(+) or pET/*Npu* DnaE/entolimod-28a(+). (C) The fusion protein sequence of *Npu* $I_C$—entolimod—*Npu* $I_N$. The C-terminal 39-residue segment and the N-terminal 102-residue segment of the *Npu* DnaE intein are enclosed within rectangles. The linker sequence of CFNHMWSHPQFEK (StrepII-tag, green type) and $H_6$ ($H_6$-tag) GS is shown in bold font. The amino acid sequence of entolimod is shown in blue type, the sequence (red type) between N- and C-terminal encodes a flexible linker domain which is replaced by a hyper-variable domain of flagellin.

the PTS reaction so far ($t_{1/2}$ of ~60 s), which was 33–170-fold higher than that of *Ssp* intein (*Zettler, Schütz & Mootz, 2009*). Thus, we could successfully express circ-entolimod by *Npu* intein-mediated its cyclization.

## Purification and thermo stability of lin- and circ-entolimod

In this step, we chosed different strategies to purify lin- and circ-entolimod, respectively. After analyzing the purity of lin-entolimod (Ni-chelating affinity) by SDS–PAGE, we only found a few bands below lin-entolimod (Fig. 3A). However, the one-step elution was not suitable for circ-entolimod purification. Therefore, we increased the concentration of imidazole (0–500 mM) in a step-wise manner to elute circ-entolimod. Figure 3B shows the result of using three imidazole gradients (50, 200, and 500 mM) to elute circ-entolimod. Unfortunately, it could not yield the same purity as lin-entolimod. We further purified lin-entolimod according to the procedure described previously, and performed desalting followed by size-exclusion chromatography (*Burdelya et al., 2008*). Through these two purification steps, lin-entolimod was purified to >95% (Fig. 3C). However, size-exclusion chromatography to purify circ-entolimod
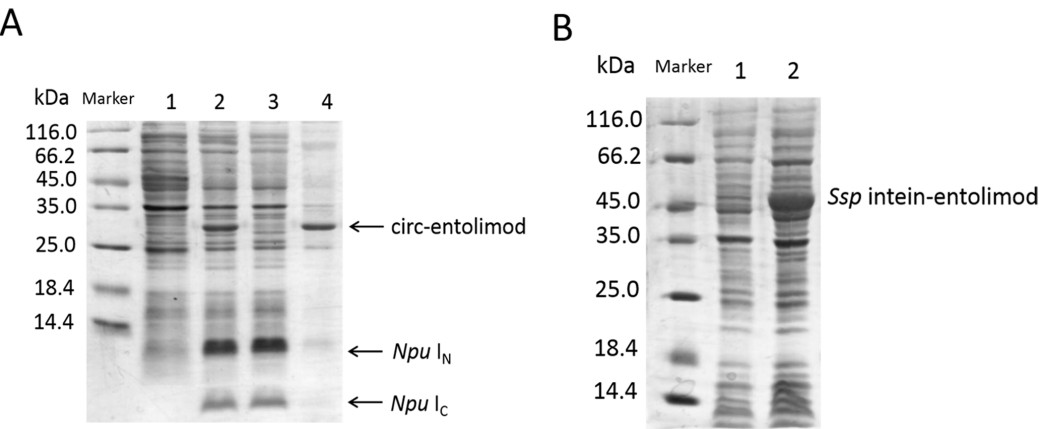

**Figure 2 SDS–PAGE analysis of circ-entolimod expression.** (A) circ-entolimod is mainly expressed as inclusion bodies. Lane 1: Before induction; Lane 2: circ-entolimod expression after induction with IPTG; Lane 3: Supernatant proteins containing circ-entolimod after sonication; Lane 4: Precipitated proteins containing circ-entolimod after sonication. The two fragments ($I_N$ and $I_C$) of *Npu* intein are clearly seen in Lane 2 and 3. (B) *Ssp* intein in mediating entolimod cyclization. Lane 1: Before induction; Lane 2: Precursor protein *Ssp* intein-entolimod after induction with IPTG.

yielded results similar to those obtained using Ni-chelating affinity purification. As predicted by an online protein isoelectric point calculator tool, the isoelectric point of circ-entolimod was approximately 5.6. We, therefore, decided to purify circ-entolimod by anion exchange chromatography using a HiTrap Q HP resin. As shown in Fig. 3D, circ-entolimod with high purity was obtained with the elution buffer containing 10 and 20 mM NaCl (lane 3 and 4). These two were then combined as the final circ-entolimod purification (Fig. 3E). To further confirm the purified product was circ-entolimod, we used MALDI mass spectrometry and Q Exactive MS to analyze its molecular weight and sequence, the results indicated that both size and sequence were right as expected (Fig. 4 and Fig. S2). In our study, at various temperatures (from 90 to 100 °C), the linear and circular form of entolimod were both performed the enhanced thermostability (Fig. S3).

## Effect of circ-entolimod on NF-κB pathway activation

To investigate whether circ-entolimod has in vitro biological activity, we added both the lin-entolimod and circ-entolimod to the dual-luciferase reporter plasmids transfected HEK293 cells. Then the luciferase activity was measured to check/determine the in vitro activity that the both products activate the NF-κB-dependent pathway by inducing production of numerous bioactive factors including anti-apoptotic proteins, reactive oxygen species (ROS) scavengers, cytokines, and anti-inflammatory agents (*Burdelya et al., 2013*; *Lockless & Muir, 2009*).

After purification, we measured and controlled the contamination levels of less than 5 endotoxin units, which was approximately equivalent to 0.5 ng of *E. coli* Lipopolysaccharides (LPS) per microgram of recombinant protein. Based on this level, LPS had minimal impact on the following detection, and then we compared the biological

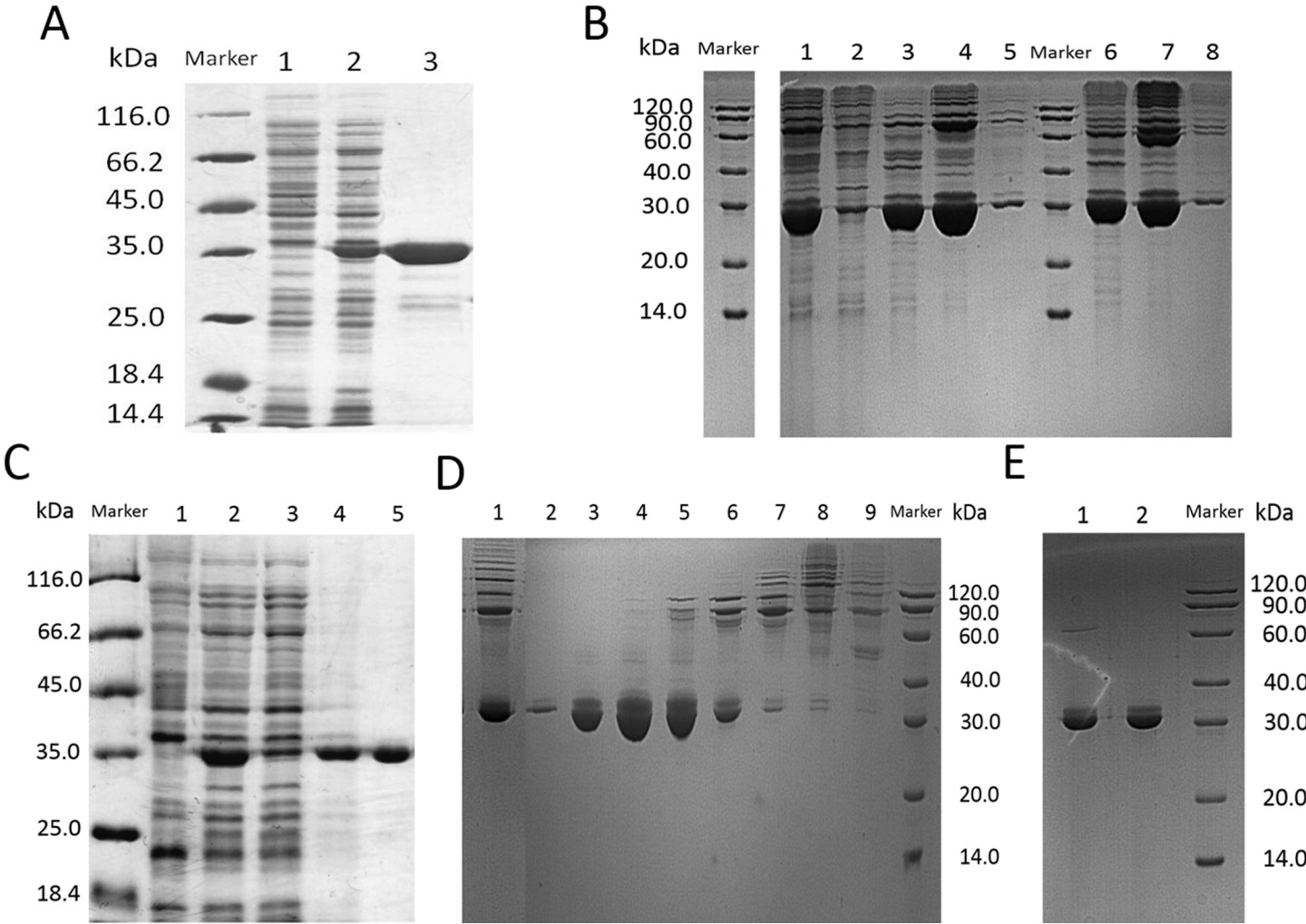

**Figure 3 SDS–PAGE analysis of lin- and circ-entolimod purification.** (A) Purification of lin-entolimod by Ni-chelating affinity chromatography. Lane 1: before induction; Lane 2: lin-entolimod expression after induction with IPTG; Lane 3: elute with 250 mM Imidazole. (B) Purification of circ-entolimod by Ni-chelating affinity chromatography. Lane 1: loaded sample; Lane 2: flow-through; Lane 3: elute with 50 mM Imidazole; Lane 4: elute with 200 mM Imidazole; Lane 5: elute with 500 mM Imidazole; Lane 6: elute with 50 mM Imidazole (Non-reduced); Lane 7: elute with 200 mM Imidazole (non-reduced); Lane 8: elute with 500 mM Imidazole (Non-reduced). (C) Purification of lin-entolimod after a second step by size-exclusion chromatography. Lane 1: before induction; Lane 2: lin-entolimod expression after induction with IPTG; Lane 3: supernatant proteins containing lin-entolimod after sonication; Lane 4: precipitated proteins containing lin-entolimod after sonication; Lane 5: final protein of lin-entolimod. (D) Purification of circ-entolimod by anion exchange chromatography using HiTrap Q HP resin. Lane 1: loaded sample; Lane 2: flow-through; Lane 3: elute with 10 mM NaCl; Lane 4: Elute with 20 mM NaCl; Lane 5: elute with 30 mM NaCl; Lane 6: elute with 40 mM NaCl; Lane 7: elute with 50 mM NaCl; Lane 8: elute with 100 mM NaCl; Lane 9: elute with 1 M NaCl. Lane 3 and 4 are combined for the final product. (E) Final protein of circ-entolimod. Lane 1: circ-entolimod (non-reduced); Lane 2: circ-entolimod.

activity of lin-entolimod and circ-entolimod at three different concentrations (0.00001, 0.001, and 0.1 μM). A total of three independent replicates were set up for each concentration and PBS control. The results shown that circ-entolimod exhibited an excellent performance at all three concentration levels. The differences between the corresponding lin-entolimod and circ-entolimod groups were statistically significant (Fig. 5, **$P < 0.01$). Additionally, even at the very low concentration (0.00001 μM), circ-entolimod still activated NF-κB signaling at approximately 40% higher than that by

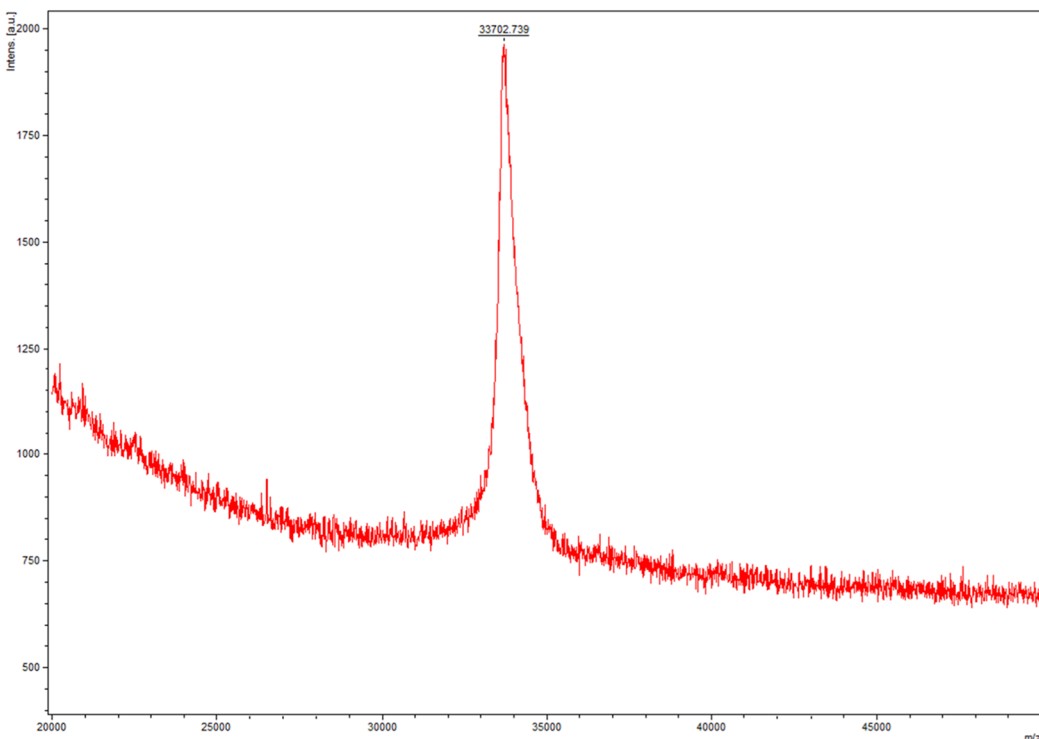

**Figure 4 The molecular weight of circ-entolimod is analyzed by MALDI mass spectrometry.** The data analysis are the means ± SD of three independent experiments. The size is ~33.7 kDa.

lin-entolimod (Fig. 5). Cyclization of proteins has been previously reported to enhance biological activity (*Iwai & Plückthun, 1999*; *Zhao et al., 2010*; *Qi & Xiong, 2017*). In the present work, our novel protein circ-entolimod also showed a similar outcome. These results thus indicate that circ-entolimod has a significant biological activity in NF-κB pathway activation.

### Effect of circ-entolimod on mouse survival after irradiation

As an anti-radiation drug, lin-entolimod showed a significant radioprotective effect on mouse and primate models. To address whether circ-entolimod has a in vivo protective effect same as or even better than that of lin-entolimod, we injected both agents s.c. into C57BL/6 mice at 0.2 mg/kg of body weight. We administered lin-entolimod and circ-entolimod 30 min prior to irradiation at two different doses (9 and 14 Gy), and compared them with control mice that received PBS (12 animals per group, in total 72 mice). We determined the survival rate of mice at 30 days after TBI at radiation doses of 9 and 14 Gy. At the lower radiation dose of 9 Gy, which caused lethality to all the PBS-treated mice, there was no significant difference in survival rate between circ-entolimod and lin-entolimod-treated mice (Fig. 6A, $P = 0.3173$). However, in the higher radiation dose of 14 Gy group, 40% of the circ-entolimod-treated mice survived whereas all lin-entolimod-treated mice were caused lethality by the ninth day (Fig. 6B, **$P < 0.01$). These results indicate that circ-entolimod has a better radioprotective effect than lin-entolimod at

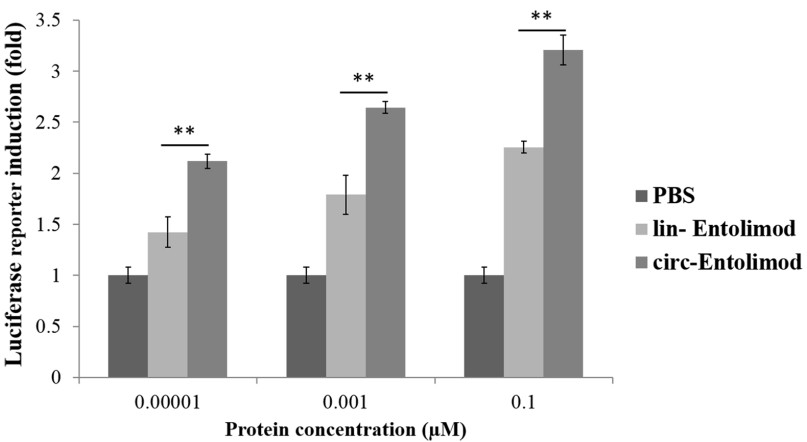

**Figure 5 Induction of NF-κB-responsive transcription by lin-entolimod and circ-entolimod.** HEK293 cells carrying an NF-κB-dependent luciferase reporter construct were incubated with the indicated concentrations (0.00001, 0.001 and 0.1 μM) of lin-entolimod or circ-entolimod. Luciferase activity was measured after 6 h. Each column represents the average of three independent experiments and error bars indicate standard deviations (S.D.). *P*-values were determined by Student's *t*-test, $^{**}P < 0.01$.

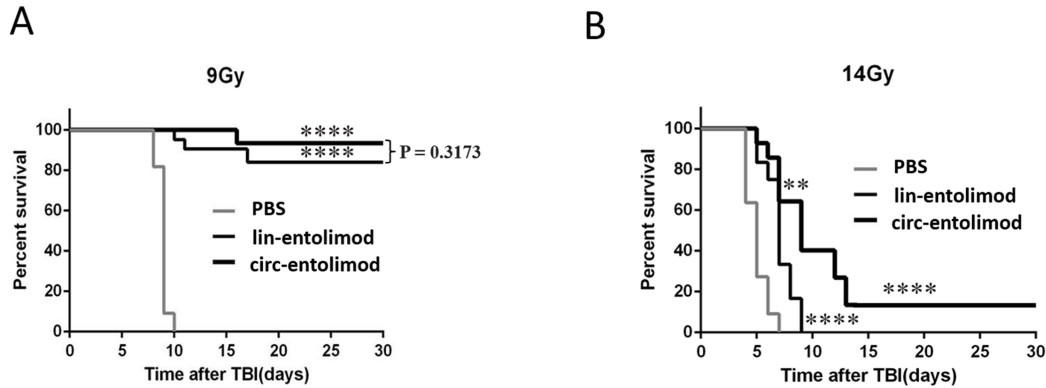

**Figure 6 Protection of mice from lethal irradiation by circ-entolimod compared to lin-entolimod.** (A) C57BL/6 mice were injected s.c. with circ-entolimod (0.2 mg/kg), lin-entolimod (0.2 mg/kg), or PBS 30 min prior to receiving 9 Gy TBI. *P*-values were determined by the log-rank test. $^{****}P < 0.0001$ for comparison of survival in circ-entolimod- and PBS-treated groups or lin-entolimod- and PBS-treated groups. $P = 0.3173$ for comparison of survival in circ-entolimod- and lin-entolimod-treated groups. (B) C57BL/6 mice were injected s.c. with circ-entolimod (0.2 mg/kg), lin-entolimod (0.2 mg/kg) or PBS 30 min prior to receiving 14 Gy TBI. *P*-values were determined by the log-rank test. $^{****}P < 0.0001$ for comparison of survival in circ-entolimod- and PBS-treated groups or lin-entolimod- and PBS-treated groups. $^{**}P < 0.01$ for comparison of survival in circ-entolimod- and lin-entolimod-treated groups.

higher radiation doses. Thus, circ-entolimod is a potential novel and effective drug with radioprotective application.

## DISCUSSION

In this study, we demonstrated that *Npu* DnaE split intein possesses robust trans-splicing activity for circularization entolimod in our SICLOPPS construct. After expression and purification, the cyclized entolimod was clearly observed upon SDS–PAGE analysis

and no precursor protein was observed, suggesting almost 100% splicing efficiency. To investigate whether circ-entolimod has better in vitro biological activity and in vivo radioprotective activity than lin-entolimod, we used the luciferase reporter assay and TBI, respectively. It was found circ-entolimod could dramatically augment NF-κB signaling activity in HEK293 cells and increase the survival rate of mice at 30 days after TBI.

In our study, the first requirement was to find a split DnaE intein with the highest splicing activity for cyclization of the entolimod. We chose *Npu* DnaE split intein for our experiment, which is significantly more tolerant of amino acid diversity in the extein sequence, as reported previously (*Townend & Tavassoli, 2016*). In intein-mediated protein, many important factors can influence the intein splicing efficiency. Previous studies indicated that a C-terminal Cys-Phe-Asn sequence was necessary to achieve highly efficient and rapid splicing (*Lockless & Muir, 2009*; *Shah et al., 2012*), but replacement of Phe with Trp or Met in the *Npu* intein also resulted in a splicing efficiency similar to that of the native extein, and the natural extein was not found to be the fastest splicing substrate (*Cheriyan et al., 2013*). *Iwai et al. (2006)* compared the co-expression of the $Npu_{I_N}/Ssp_{I_C}$ with that of $Ssp_{I_N}/Ssp_{I_C}$ constructs. They found that the extein sequence at the splicing junction has little importance for the splicing activity. Another important reason for the higher activity of *Npu* intein is possibly the substitution of less conserved residues. The key residues in the active sites of both inteins may slightly alter their own geometry or polarization, which might influence the splicing reaction (*Zettler, Schütz & Mootz, 2009*). Together, we speculated that the requirements for trans-splicing seem to be specific to the inteins themselves, and that the difference in splicing efficiency between *Npu* and *Ssp* inteins may arise from the proper formation of their three-dimensional structures.

Besides, as the lin- and circ-entolimod have different topological structure, the purification strategies for lin- and circ-entolimod are different. After purification by Ni-chelating affinity chromatography, we found that the distribution of *E. coli* host proteins was quite different between the one-step purified products of lin-entolimod and circ-entolimod. As shown in Figs. 3A and 3B, the extra non-target bands mainly migrated below lin-entolimod contrasted to above circ-entolimod. We assumed that the process of intein-mediated entolimod cyclization might have caused this different, because the circularized circ-entolimod product might migrate faster relative to the liner one. This expectation was supported by the following purification procedures where we could obtain high purity for circ-entolimod by IEC but not by size-exclusion chromatography.

The biological safety of lin-entolimod has been tested in 150 healthy human subjects. A phase I study of lin-entolimod in patients with advanced solid tumors has been completed and a second one is now undergoing (http://www.cbiolabs.com/entolimod). Additionally, a phase I clinical trial of lin-entolimod in patients with metastatic liver disease is under preparation (*Brackett et al., 2016*). However, because entolimod is a flagellin variant, it can still cause a "flu-like" syndrome after administration. Therefore, it is essential to further reduce its potential immunogenicity and toxicity. The circular protein circ-entolimod has shown significant much better in vitro and in vivo biological activity than the linear one. Thus, the same treatment effect could be obtained by using reduced doses of circ-entolimod

than the linear one, which may reduce the potential immunogenicity and toxicity. Moreover, protein head-to-tail cyclization is not susceptible to hydrolysis by exogenous proteases, thus might be a general approach to increase the stability of linear proteins (*Iwai & Plückthun, 1999*; *Iwai, Lingel & Pluckthun, 2001*; *Zhao et al., 2010*; *Qi & Xiong, 2017*). Following the above principle to circ-entolimod, the time frame for effective administration would be widened and could improve its therapeutic effect.

## CONCLUSION

The above results indicate that circ-entolimod might be a good candidate for further clinical investigation.

## ACKNOWLEDGEMENTS

We thank members of our laboratory for critical discussions.

### Funding

This research was supported by the National Basic Research Program of China (2013CB966802), National Natural Science Foundation of China (31370762, 31030026, 31071119, 81372218) and National Key Technology R&D Program of China (2012BAI01B07). The funders had no role in study design, data collection and analysis, decision to publish, or preparation of the manuscript.

### Grant Disclosures

The following grant information was disclosed by the authors:
National Basic Research Program of China: 2013CB966802.
National Natural Science Foundation of China: 31370762, 31030026, 31071119, 81372218.
National Key Technology R&D Program of China: 2012BAI01B07.

### Competing Interests

The authors declare that they have no competing interests.

### Author Contributions

- Bingyu Ye conceived and designed the experiments, performed the experiments, analyzed the data, prepared figures and/or tables, authored or reviewed drafts of the paper, approved the final draft.
- Wenlong Shen analyzed the data.
- Minglei Shi contributed reagents/materials/analysis tools.
- Yan Zhang contributed reagents/materials/analysis tools.
- Cunshuan Xu conceived and designed the experiments, contributed reagents/materials/analysis tools.
- Zhihu Zhao conceived and designed the experiments, prepared figures and/or tables, authored or reviewed drafts of the paper, approved the final draft.

## Animal Ethics

The following information was supplied relating to ethical approvals (i.e., approving body and any reference numbers):

All animal experiments were performed according to the Guide for the Care and Use of Medical Laboratory Animals (Ministry of Health, China, 1998) and with the ethical approval of the Beijing Institute of Biotechnology (No. 545 [2001]).

## Data Availability

The raw data are provided in the Supplemental Files.

## Supplemental Information

Supplemental information for this article can be found online at http://dx.doi.org/10.7717/peerj.5043#supplemental-information.

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
