# Peer review of "Intein-mediated backbone cyclization of entolimod confers enhanced radioprotective activity in mouse models"

_PeerJ, doi:10.7717/peerj.5043_

## Round 0.1 · original submission · Major Revisions

Please address all critical points raised by the reviewers and revise the manuscript accordingly.

Reviewer 1 ·

Basic reporting

Ye et al. report that backbone cyclization of a truncated flaggelin protein, termed Entolimod, using a split intein from Nostoc punctiforme and showed that the circular form of entolimod has increased in vivo stability and activity than the linear form. This could be hugely advantageous for the clinical application of entolimod, which has anti-cancer activity as well as radioprotective activity. The article is written logically and understandable, but could still be polished by indicating the important information more clearly. There are several technical issues in the manuscript, which needs to be addressed (see below).

Experimental design

The authors used the split intein to create circularized entolimod to test in vivo activity for the comparison with linear form and produce the circular form by head-to-tail backbone cyclization using protein trans-splicing as well as non-circularized liner entolimod for the comparison. The both forms were used for reporter assays and mice survival experiments to demonstrate the in vivo stability. The authors claim that C-entolimod was purified from the inclusion bodies but no solubilizing conditions is described. It is very unclear how the proteins are purified. This is the crucial point of the experiments. Moreover, the sequence as entolimod provided by the authors in Fig.1 contains additional StepII-tag and His-tag.
To my knowledge, no protein with His-tag has been currently approved by FDA. Does entolimod currently under development as drug contain the identical sequence?

Validity of the findings

Typically, backbone circularized proteins migrate faster than the liner form in SDS-PAGE, when N-and C-termini are in the proximity, which is not clear from the manuscript. The migration in SDS-PAGE has been used for confirming backbone cyclization because it could migrate slower (increased apparent molecular mass) after proteolytic digestion, which is unusual. However, Fig.3E shows the identical migration in SDS-PAGE for both forms. This is very unusual for the backbone cyclization. There are also two bands in Fig. 3E. What is the other band? The authors could use different percentages of acrylamide SDS-gel for better separation.
In Fig. 4, the expected molecular weight can be indicated for the clarification. Is this MALDI-TOF spectrum? In the experimental section, it is written as QE-Mass? Is this spectra convoluted or the original? Why the authors modified cycteins with IAA. 18 Dalton difference is expected for the difference between linear and circular forms. Therefore, the mass spectrum should be obtained by an appropriate mass spectrometer for ionizing and detecting.

Additional comments

1. The authors used “L-entolimod” and “C-entolimod” for linear form and Circular form respectively. This nomenclature could be confusion as “L-form” is often used for indicating the chirality. I recommend to use “l-“ and “c-“ in italic, or “lin-“ and “circ-“.
2. Line 141, the authors wrote“L-entolimod was purified described previously (inclusion bodies did not require refolding)(Burdelya et al., 2008)”. I think it is not inclusion body if the protein does not require any refolding but soluble.
3. Line 93, the authors wrote “C-entolimod is its increased stability”. This is not true.
The authors provided only in vivo stability without anlazying thermodynamical stability. Why didn’t the authors analyze and compare the thermostability? This is a very easy and quick experiment to perform.
4. The sequence in Fig. 1 should be presented in a clear manner. There are His-tag and StrepII-tag sequences, which, I believe, is not the part of entolimod. The authors could present the sequence of entolimod clearer, indicating the part of the flagellin removed.
5. Section 2.1. in method and material could be simplified so that one could follow them easily.
6. Circular GFP and increased stability was originally reported in 2001, which should be cited (J. Biol. Chem. 276, 16548-54)
7. The authors cite several addgene plasmid numbers but did not include the references for these plasmids, which are supposed to be cited.
8. Line 213, Townend and Tavassoli, 2016 is incorrect reference for this.
9. Did the authors try to express at a lower temperature to increase soluble proteins? It is mysterious how the protein could be purified without any denaturant from inclusion bodies.
10. In my opinion, SCICLOPPS is a bad jargon created.

Reviewer 2 ·

Basic reporting

The manuscript is written in clear professional language and technically correct. The introduction and background are sufficient to identify a problem, emphasize the significance of the work and impact on the field. The manuscript is supported by a sufficient number of references possessing both quality and relevance. However, in the introduction and also the abstract, the authors write about the flu-like syndrome induced by entolimod (lines 20 and 48), but the references in the introduction do not describe these observations which most likely were drawn from Phase I safety clinical trials. If entolimod was indeed used in the clinic for radioprotective and anticancer clinical application when the “flu-like syndrome” was observed, appropriate references must be provided. In both the abstract (line 24) and the introduction (line 86), the authors state that different strategies were used to purify linear and cyclized entolimod. The methods of purification should be provided and described or not mentioned at all if they are unimportant.
Overall, the structure of the article is very good with complete presentation of material that is sufficiently described and supported by raw data. The authors sought to create a circular version of entolimod hoping to improve its biological properties. The results of the study support the hypothesis.

Experimental design

The experimental work was conducted with rigor and to a high technical standard. There is concern that C-entolimod could contain LPS from the E.coli purification. As the authors show that C-entolimod is similar or better then linear entolimod with regard to NF-кB activation and also radioprotection, it is important to confirm that it is mediated by TLR5 and not TLR4. One possible approach to do this is to add data showing that C-entolimod does not activate NF-кB in the cells expressing TLR4, but not TLR5.

Validity of the findings

The authors clearly present their conclusions and validate them with their supporting results. This is an interesting approach to increase the activity of entolimod while reducing its undesirable properties. Although the degree of improvement in tissue protection and reduction in toxicity and immunogenicity must now be determined for this new variant of entolimod, the current manuscript describes the steps to obtain a cyclic version of entolimod and shows the increased NF-кB activating capacity as compared to the parental linear version. The existence of a cyclic version of entolimod with improved tissue protecting capacity would enable further investigation of the mechanisms responsible for this activity and possible development of the drug for prevention of radiation toxicity in hematopoietic and gastrointestinal tissues.

Additional comments

Minor issues:
1. In figure 6, statistical differences are described in the figure legend, but not shown in the plots making up the figure.
2. lines 274 and 277: it would be more appropriate to use term “cause or prevent lethality” rather than “killed”
3. line 133: remove “an”

---

## Round 0.2 · accepted · Accept

Thank you for carefully addressing critiques of both reviewers and for adequate revision of the manuscript. Since all the concerns were addressed, the revised manuscript is accepted for publication in PeerJ.

#